# Impacts of clinical academic activity: qualitative interviews with healthcare managers and research-active nurses, midwives, allied health professionals and pharmacists

Lisa Newington ,[1,2,3] Caroline M Alexander ,[1,2] Mary Wells [1,3]

¹Surgery and Cancer, Imperial College London Faculty of Medicine, London, UK
²Therapies, Imperial College Healthcare NHS Trust, London, UK
³Nursing Directorate, Imperial College Healthcare NHS Trust, London, UK

**Correspondence to**
Dr Lisa Newington;
l.newington@imperial.ac.uk

## ABSTRACT

**Objectives** To explore the perceived impacts of clinical academic activity among the professions outside medicine.

**Design** Qualitative semistructured interviews.

**Setting and participants** There were two groups of interviewees: Research-active nurses, midwives, allied health professionals, healthcare scientists, psychologists and pharmacists (NMAHPPs) and managers of these professions. All participants were employed in a single, multisite healthcare organisation in the UK.

**Analysis** Interview transcripts were analysed using the framework method to identify key themes, subthemes and areas of divergence.

**Results** Four themes were identified. The first, cultural shifts, described the perceived improvements in the approach to patient care and research culture that were associated with clinical academic activity. The second theme explored visibility and included the positive reputation that clinical academics were identified as bringing to the organisation in contrast with perceived levels of invisibility and inaccessibility of these roles. The third theme identified the impacts of the clinical academic pathways, including the precarity of these roles. The final theme explored making impact tangible, and described interviewees' suggestions of possible methods to record and demonstrate impact.

**Conclusions** Perceived positive impacts of NMAHPP clinical academic activity focused on interlinked positive changes for patients and clinical teams. This included delivery of evidence-based healthcare, patient involvement in clinical decision making and improved staff recruitment and retention. However, the positive impacts of clinical academic activity often centred around individual clinicians and did not necessarily translate throughout the organisation. The current clinical academic pathway was identified as causing tension between the perceived value of clinical academic activity and the need to find sufficient staffing to cover clinical services.

## INTRODUCTION

It is widely reported that healthcare organisations that engage in clinical research have better outcomes than their non-research active counterparts.[1–5] For example, research

### STRENGTHS AND LIMITATIONS OF THIS STUDY

⇒ This qualitative evaluation illustrates research impacts from the perspective of research-active clinicians and healthcare managers in the professions outside medicine.
⇒ Existing methodological frameworks for the assessment of research impact focus at the organisational or national levels, whereas we provide individual perspectives.
⇒ The study was limited to employees at a single healthcare organisation and may not reflect other settings.

activity has been associated with improvements in organisational performance and efficiency, patient satisfaction and confidence in their healthcare professionals and staff satisfaction.[3 6] Research activity has also been associated with reductions in mortality and staff turnover.[2 5 6] Consequently, the UK Care Quality Commission Well-Led inspection framework now includes specific assessment of clinical research activity and leadership.[7] A number of frameworks have been developed to aid recording of research impact both within and across organisations.[8 9] These have largely focused on academic metrics, such as publications, citations and securing further funding. However, the pertinent components of research impact vary across different contexts,[10 11] and may include other aspects that are not traditionally measured or recorded. Our recent systematic review used a modified VICTOR framework (making Visible the ImpaCt Of Research) to classify the reported impacts of healthcare research led by clinicians from outside medicine.[12 13] This included broad categories of impact, such as: economic; knowledge exchange; service provision and workforce; and research profile, culture and capacity. It also incorporated the

individuals who might be affected: patients; staff (recruitment/retention) and clinical academics. Across these domains, there were several recurring elements that illustrated the challenges and benefits of balancing clinical and academic roles, the creation and implementation of new evidence, and the development of collaborations and networks.

Within medicine, there are various career pathways and structures to support clinical academic roles.[14 15] Opportunities for non-medical clinicians to engage in research alongside their clinical practice are now increasing, particularly through schemes such as the National Institute for Healthcare Research and Health Education England funded 'Integrated Clinical Academic' Fellowships.[16] This is in addition to research leadership capacity building initiatives such as the NHS (National Health Service) 70@70,[17] research internships for newly qualified clinicians,[18] nursing, midwifery and allied health professionals research awards,[19] and discipline-specific research capacity building initiatives, such as the NIHR (National Institute for Health Research) Nursing and Midwifery Incubator.[20]

Imperial College Healthcare NHS Trust (a large hospital group within the UK National Health Service) has developed a strategic plan to increase and support clinical academic activity among the professions outside medicine.[21] We initiated a qualitative interview study to explore individual perceptions of the impacts of this clinical academic activity, and understand any differences between the views of managers and research-active clinicians. An additional component of this study explored the question of 'what is a clinical academic?', and has been reported elsewhere.[22]

## METHODS
### Design and approvals
The study was approved by the Imperial College Healthcare NHS Trust Clinical Audit Team (reference: 418) and followed a prespecified protocol.[23] Additional NHS ethics approval was not required.[24] The research team comprised postdoctoral clinicians from nursing and physiotherapy disciplines with previous qualitative research experience. Qualitative semistructured 1:1 interviews were conducted using prepiloted topic guides which were informed by our systematic review of the literature[13] (online supplemental file 1). The COREQ checklist (COnsolidated criteria for REporting Qualitative research) was used to guide reporting.[25]

### Patient and public involvement
The focus of this study was on understanding the perceptions of healthcare managers and clinicians from the professions outside medicine. Patient/public advisors were not specifically involved; however, the wider topic of research impact was discussed with two public representatives as part of our larger programme of research.[26] Interview topic guides were developed in collaboration with

research-active clinicians from both within and outside the NHS Trust. Involvement included providing feedback and suggestions on the initial draft, piloting and further refining the final version. The medical and dentistry community were not included in this stakeholder work because their clinical academic careers are already well established, and our emphasis was solely on the professions outside medicine.

### Participants and recruitment
Imperial College Healthcare NHS Trust is a large multi-site NHS organisation situated in north west London, UK. The Trust provides a range of specialist healthcare services located in both inpatient and outpatient settings, and serves around a million people per year, with a staff of >13 000.[27] Eligible research-active clinicians were healthcare professionals from any discipline outside medicine who worked within the NHS Trust and were engaged in clinical academic activity.[28] This included: nursing; midwifery; the allied health professions (art therapy, dietetics, drama therapy, music therapy, occupational therapy, orthoptics, operating department practitioners, osteopathy, podiatry, prosthetics and orthotics, paramedics, physiotherapy, radiography, and speech and language therapy); healthcare science, psychology and pharmacy, as abbreviated to NMAHPPs. Clinical academic activity was defined as engagement in research alongside clinical practice that was supported by additional funding from clinical research organisations or charities. This included both full and part-time research secondments.

Eligible managers were those responsible for managing any of the professional groups described above. This ranged from line managers through to higher level service managers. Permission to directly contact individuals was granted through the Trust Clinical Audit and Service Evaluation Team. Potential interviewees were contacted by email, using addresses that were openly accessible through the NHS or university email systems.

A purposive sampling strategy was adopted to ensure inclusion of a range of experiences. This also included snowball sampling techniques, with interviewees and potential interviewees asked to suggest other research-active clinicians and managers. Sampling criteria and recruitment processes are outlined in table 1. All participants provided informed written consent after reviewing the participant information sheet (online supplemental file 2). Interviews were conducted by the lead researcher and were delivered face to face or remotely, according to interviewee preference. The interviews were audio recorded and transcribed verbatim by an external transcription company bound by a non-disclosure agreement. Transcripts were anonymised and returned to participants for comment/correction. Anonymisation included removal of names, clinical disciplines, locations and other potentially identifiable characteristics. Recruitment continued until the research team were confident that data saturation had been achieved and the purposive sampling criteria were met. Saturation was defined

**Table 1** Recruitment details and sampling criteria

| | Research-active clinicians | Managers |
|---|---|---|
| Identification | 1. Existing database of healthcare professionals at the Trust with external research funding<br>2. Open invitation via Trust Twitter and e-bulletin | 1. Trust leadership directory<br>2. Open invitation via Trust Twitter and e-bulletin<br>3. Suggestions from interviewees |
| Recruitment | 17 email invitations<br>13 agreed to be interviewed | 11 email invitations<br>9 agreed to be interviewed |
| Sampling criteria | Clinical discipline and/or specialty | Clinical discipline and/or specialty |
| | NHS (National Health Service) grade | NHS grade |
| | Gender | Gender |
| | Hospital site within the Trust | |
| | Academic level | |

as the interviewer hearing the same or similar content, and when no new codes were identified during data analysis.[29 30]

## Analysis

Data were managed and analysed using the Framework Method,[31 32] supported by NVivo V.12 software (QRS International). The authors independently coded the first two transcripts and agreed the preliminary coding framework, which was applied to all transcripts by the lead author. Codes were added and modified in response to newly identified items. Any changes were agreed by all authors, and retrospectively applied to precoded transcripts. Coded text was summarised, and analytical ideas were logged and explored thematically by all authors, using the NVivo framework matrices function, to identify recurring and unique themes discussed by interviewees. Preliminary themes and subthemes were shared with all the interviewees, nine of whom provided feedback that was incorporated into the final findings (six research-active clinicians and three managers). In addition, preliminary findings were presented to the Trust Postgraduate Research Forum (research-active clinicians from non-medical disciplines) for feedback and comment. Example feedback included: discussion on how the individual codes could be arranged as themes and subthemes; suggestions for the wording of the theme headings and ideas for the design of the summary model.

## RESULTS

Twenty interviews took place between February and July 2020 (12 research-active clinicians and 8 managers). Participant demographics and interview details are provided in table 2. None of the invited managers or research active clinicians actively refused to participate, but there were two non-responders within the manager group, and four within the research-active clinician group. An additional two individuals (one in each group) were unable to schedule an interview due to changeable work commitments associated with the

COVID-19 response. Three individuals responded to the Twitter invitation, although all three had already been identified as potential participants. All purposive sampling criteria were met, with the exception of gender. However, the predominance of women reflects both the local and international distribution of non-medical healthcare professionals.[33]

Three non-hierarchical and interlinking themes were developed that described the reported impacts of clinical academic activity (figure 1). The first theme explored perceived cultural shifts that both involved and extended beyond individual research-active clinicians. The second theme described diverging levels of visibility for the research-active clinicians within different settings. The third theme examined the challenges and opportunities of the existing clinical academic pathways. In addition, a final theme explored making impact tangible and described interviewees' suggestions of possible methods of capturing impact. All themes are described below with illustrative quotes, and additional quotes are provided in table 3. No themes or subthemes were specific to either managers or research-active clinicians and any unique or diverging views among individuals were explored within the subthemes.

## Cultural shifts

Clinical academic activity was perceived to contribute to beneficial cultural changes relating to the provision and delivery of clinical care and research engagement. Many research-active clinicians recalled how they had noticed positive changes in their approach to patient care, which were also adopted by other team members. In addition, managers named clinical academics within their teams as exemplars, highlighting the positive contributions they were making to the local research culture.

### Approach to patient care

Reported changes to patient care were not isolated to the implementation of findings from the research-active clinicians' own research. Perceived impacts included: increased confidence in questioning practice and openly discussing with patients and colleagues

**Table 2** Participant demographics and data collection details

| | | Research-active clinicians | | Managers | |
|---|---|---|---|---|---|
| Participant details | Interviewees | 12 | | 8 | |
| | Clinical discipline | Nursing | 4 | Nursing/midwifery | 3 |
| | | Midwifery | 2 | Allied health professions | 3 |
| | | Speech and language therapy | 2 | Pharmacy | 1 |
| | | Occupational therapy | 1 | Multidisciplinary | 1 |
| | | Radiography | 1 | | |
| | | Dietetics | 1 | | |
| | | Pharmacy | 1 | | |
| | Gender | Female | 10 | Female | 7 |
| | | Male | 2 | Male | 1 |
| | Hospital site | A | 3 | Multisite | 8 |
| | | B | 2 | | |
| | | C | 4 | | |
| | | D | 3 | | |
| | Date of clinical qualification | Median 2004 Range 1984–2016 | | Not collected | |
| | Academic level | Predoctoral | 5 | Not collected | |
| | | Doctoral | 3 | | |
| | | Postdoctoral | 4 | | |
| Data collection | Interview format | Face to face | 3 | Face to face | 2 |
| | | Video call | 6 | Video call | 4 |
| | | Audio call | 2 | Audio call | 2 |
| | | Email | 1 | | |
| | Interview duration | Mean 57 min | | Mean 45 min | |
| | | Range 45–70 min | | Range 27–62 min | |

if there was uncertainty over management options; increased involvement of patients in evidence-based treatment decision-making; improved problem solving; and greater awareness of the burden to caregivers. These impacts were reported by both groups of interviewees. Research-active clinicians (R) reflected on their individual experiences, while managers (M) identified how research-active clinicians had generated improvement throughout their clinical team, as illustrated by R2, R6 and M1:

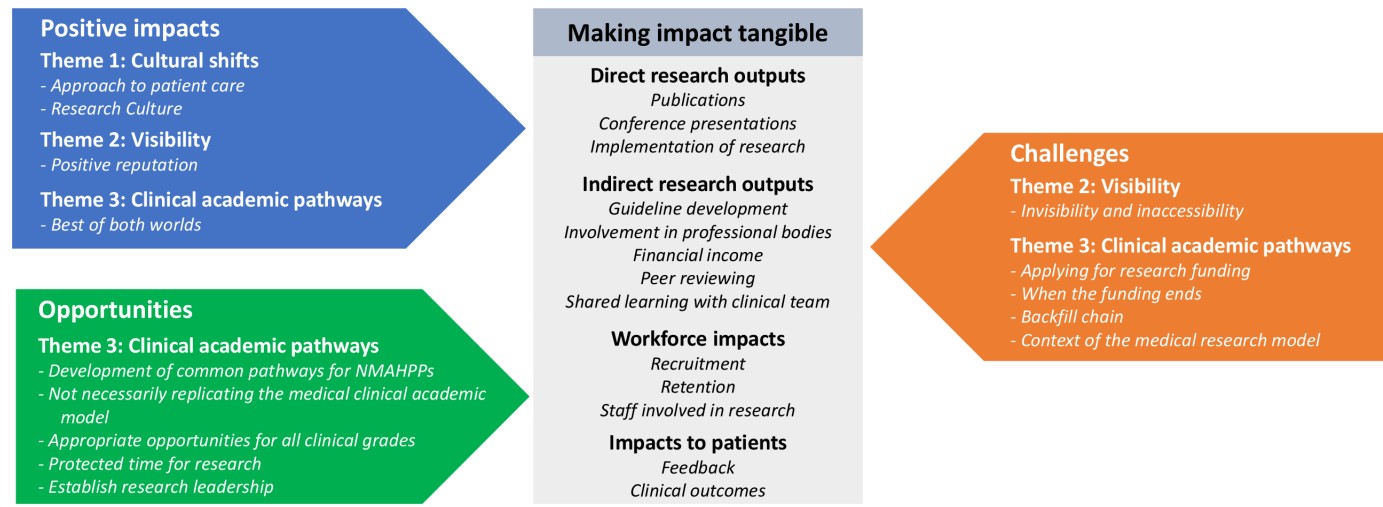

**Figure 1** Thematic representation of the impacts of clinical academic activity. NMAHPPs nurses, midwives, allied health professionals, healthcare scientists, psychologists and pharmacists.

**Table 3** Additional quotes supporting the identified themes and subthemes

| Theme | Subtheme | Illustrative quote |
|---|---|---|
| Cultural changes | Approach to patient care | *I guess anyone that's going into research is quite passionate about that, and they will often have a specific area that they are passionate about, and keen to improve, bring back any new learning or innovations from outside into clinical practice.* M8<br>*You really start to appreciate how much we overlook the role and the burden and influence that caregivers have, and whilst we've always tried to obviously include them in our clinical work before, all of our clinical work has a focus which is the patients.* R12, pre-doctoral |
| | Research culture | *Support from more experienced clinical academics who can help guide through the challenges that are often faced in this role such as timelines, funding, bureaucracy…* R10, post-doctoral<br>*There weren't many role models in the UK… We need more professors, more AHP professors; there's actually very few at the moment.* R6, post-doctoral<br>*Well, their enthusiasm, for a start, and they can, you know, bring the team with them with their research.* M6 |
| Visibility | Positive reputation | *I think the PR [public relations], you know, [named research-active clinician] is constantly on Twitter, and so it's using the [organisation's] name, the papers that we get published, and again, it's got the [organisation's] name on it.* M6<br>*If staff are learning and being challenged, they're probably less likely to leave.* R3, post-doctoral |
| | (In)visibility and (in)accessibility | *When I came back from my [doctoral] fellowship, in many ways I almost felt as though I came back in a [junior] role, where I was just churning out patients and not really having the opportunity to share those skills that I had, or to upskill the team.* R3, post-doctoral<br>*It's about understanding that… it's not that people don't want to apply, it's that there are barriers so you're moving… you're kind of moving the onus on them not applying rather than the fact that maybe they have different barriers that we haven't considered.* R7, doctoral |
| Clinical academic pathways | Applying for research funding | *I've been looking at the most optimum funding opportunities for me to continue what I would like to do as the next stage of my research. I've been looking at the NIHR Advanced Fellowship, looking at the timelines for that, the application form, and I've been reflecting… I've really pulled out the feedback from that and looked at it very carefully to see what sort of work I need to do to make my next application successful.* R6, post-doctoral |
| | When the funding ends | *The other side of the coin is that it's difficult to progress while you're in it [research fellowship]… It's hard to progress clinically, it kind of puts you on pause, even though I feel like my clinical skills are developing.* R2, pre-doctoral<br>*We've been trying to create those roles for a few years now, but the challenge is trying to kind of get the funding to have those dedicated sessions for research, and so the only way in which we're achieving it is through grants and things like that, so all of the research activity going on at the minute is only through awards that enable us to have some research time… But it's not certain, so it's really hard to plan for that, both the clinical backfill piece, but also in terms of what you can then achieve or plan for research-wise.* M8 |
| | Backfill | *They [research-active clinicians] get some funding to do some research and then they're with us part time and then we have to backfill them and that creates an operational pressure.* M1 |
| | Medical model | *If you're a doctor on a career in a medical pathway, you will have time out to go and do your research or you'll be required to do your audit or to get your portfolio signed off. You'll need all of that. But actually we're not required to have that, and then we are so desperate just trying to find this frontline staffing.* M3<br>*We [NMAHPPs] don't have that expectation of engaging [in research], and that may need to change actually. I think a lot of the narrative we've got about clinical academics comes from the really well-defined pathways of the classic physician. I'm happy that that exists, of course, but I'm anxious that it all seems to be about that kind of model – not anxious, I'm just sometimes unhappy that it all seems to be geared towards having that same model.* R4, post-doctoral |
| | Best of both worlds | *The investment is in you to develop it, and that's brilliant, so that means there's a lot of scope to deepen your academic interest area and develop a project. I can see what needs to be done and I don't see blockages; I just see a lack of history with it.* R5, pre-doctoral |
| Making impact tangible | Direct research outputs | *I think our definition of impact needs to change radically, because so far impact is publications, and that is all that matters unfortunately… I hope – I think, and I hope, it is starting to change… because the impact that [we need to measure] or want to see is the change that happens in practice.* R9, doctoral |
| | Indirect research outputs | *So I think a lot of the staff who are research active are then the ones who sit on various, you know, we've got someone that sits on a [multi-disciplinary national clinical area professional] body, we've got various people who are considered experts within our professional body, who contribute papers and things like that. So I don't know whether that is also a measure of impact in terms of the national influence, because of the bodies that they sit on.* M8 |
| | Workforce impacts | *I feel like it's made me appreciate [the] Trust so much more, that I've got this opportunity with them and, I just feel really lucky to work here because in other places it might not have happened… Retention is a problem across [the NHS], especially in London, and I'm one of… I'm one of the longer [serving] ones!* R2, pre-doctoral |
| | Impacts to patients | *We showed that we could deliver a one stop clinic and that patients loved it, and that they could have surgery on the day, and that they don't need follow-up and they do just fine. So we showed a model that patients really liked, had great satisfaction with and great outcomes, and at the same time saved the trust money.* R3, post-doctoral<br>*It would be interesting to capture ways of working or new arrangements of leadership. You could look for evidence of that from the patients and people themselves. They're qualitative things but you can add them up.* R5, pre-doctoral |

*I feel like my standard of care has improved because I'm questioning my practice more, I'm quite reflective in my practice and I think that's because I'm trying to think of how can I improve my practice… I think it's creating this environment [in the department] of people questioning and wanting to* *improve their practice through what's current, which is really nice to see.* R2, pre-doctoral

*I suppose my research experience has allowed me to be very upfront with patients and say: 'You know what, this is the evidence we've got so far, I'm going to ask you to do these*

exercises, we think these work for some patients but we don't have enough information yet to know whether they work for all patients. With this in mind, do you still want to proceed?'. So I guess in that way it's helping me to make sure that [the] intervention I provide is more patient-led. R6 post-doctoral

If you're doing the research it does make you a better clinician in terms of your problem solving and your thinking." M1

### Research culture

The majority of research-active clinicians reported that a key personal impact of their research engagement was the opportunity to establish and develop networks with other clinicians who were also interested in research. This included both formal and informal networks, and involved individuals from a range of disciplines. Perceived benefits included being exposed to different research methodologies and research opportunities, practical guidance and becoming connected with like-minded individuals, as discussed by R5 and R7:

For example, I've gone to some weekend residential things where it's a hodgepodge of clinicians but all with the academic pathway… and you all have the same language and lens that you're doing things from… that's made it really interesting because I have developed really far-flung contacts and networks, so that's been great. R5, pre-doctoral

It introduces you to people, like you, who are doing similar things but in other places. And so you don't feel like what you're doing is like… you don't feel like you're on your own in a way… It gives you links across the UK [and] exposure to certain areas and people that I normally wouldn't get. R7, doctoral

However, several interviewees also highlighted that despite the support of different individuals and networks, there appeared to be a lack of clinical academic role models for them, particularly at a postdoctoral level. This was identified both within the organisation and nationally, as noted by R9:

I don't have any [clinical discipline] who has a PhD, there is no advanced practitioner, I don't have any [clinical discipline] who is at a higher level who could say okay I want to be your sponsor… So, I am finding it difficult to find ways of linking better with the clinical team and utilising the tools and the skills that I have been developing over time. R9, doctoral

The importance of role models was also raised by managers, who indicated that a perceived positive impact of the research-active clinicians within their teams was the provision of inspiration and support for other clinicians, as illustrated by M1 and M8:

We have some amazing people who are complete pied pipers… and we need pied pipers in the academic world and in the clinical world and in the evidence-based practice world. M1

We're quite a research-active service. We've got quite a few people who are engaged within research, and I think that kind of has bred itself. And I think it also would attract quite a few people [to work at the organisation]… Also, having the research-active staff members as well helps promote research within the teams. M8

Both groups of interviewees discussed the perceived positive impacts research-active clinicians had on a drive towards research and evidence-based practice. This included building research skills and expertise and fostering research engagement, as recalled by R6 and R2. However, some interviews identified that this appeared to be largely driven by the passion and enthusiasm of individual clinical academics, and it was unclear whether this would lead to a sustained change, as reported by M3:

It means that the research becomes part of our business as usual in terms of clinical care. And that's for us as well as for our patients. R6, post-doctoral

I think it is creating this environment [in the clinical department], which is really nice, of people questioning and wanting to improve their practice through what's current… People come in to approach me… and if they have seen a piece of research, they've talked to me about it…. and they'll ask me about it. It starts a lot of conversations. R2, pre-doctoral

I think the people who do it are passionate about wanting to see improvement. And whilst they're in a service, their passion is spread across their team. What I've realised is, if they move on, it's not always embedded. M3

### Visibility

Visibility of research-active clinicians was widely discussed, and these individuals were believed to generate a *positive reputation* for the Trust and their clinical discipline more generally. However, within the Trust setting, many interviewees perceived a lack of visibility of their research outside their immediate clinical departments. This led to an interesting discordance between the positive reputation of clinical academics coupled with (in)visibility and (in)accessibility of these roles.

### Positive reputation

The perceived positive reputation largely stemmed from showcasing clinical academic successes and opportunities. This included academic outputs, such as publications and presentations as well as developing a national standing, with individuals being contacted to provide clinical and research expertise, as summarised by R11 and M4:

On a national level being seen… people do look to me now as someone who's really taking a lead on that research and so, obviously you have people contacting you for help and support and that kind of thing that comes with it. So, yeah, just being seen as a sort of advocate for that kind of research in our patient group. R11, pre-doctoral

Having profile and contacts and a voice that carries weight, and that is supported by being a clinical academic without

*doubt. And enhances the reputation of an NHS service to have experts that are recognised internationally for their research, as well as for their clinical expertise.* M4

This positive reputation was also perceived to contribute to improved recruitment of clinical staff to the Trust, and the retention of existing staff, as illustrated by M1 and R1:

*They're [clinical academics] great for profiling us. They attract people... I don't know how many interviews I've sat in where people say 'Well I wanted to come to an AHSC [Academic Health Science Centre] and I know you've got [named clinical academic] here'... So it's good for recruitment and retention, I think it's an aspirational place to work and I think the clinical academics help us to keep it that way.* M1

*There's something around the impact of people being research active, or having clinical academics in teams, around recruitment and retention. I don't know, but are people more likely to stay in a Trust where they can see that's an opportunity for them and an option for them... [this could be] a way of keeping people and skilling them up.* R1, doctoral

### *(In)visibility and (in)accessibility*

Despite widespread reports that clinical academic activity was beneficial for the reputation of the Trust, many interviewees also reported that their research findings and expertise were underutilised and unknown outside their immediate clinical area, as recalled by M7, R9 and R4:

*I just don't think it's got the profile that it needs to have. I don't quite know how that should be improved, but proper, [clinical discipline] research, I don't think, is well understood or widely talked about or well known.* M7

*I still feel that there needs to be more showcasing of what is being done... I don't think it is reaching the people that it needs to reach. So, for example the network events or the [organisation] research hubs and all those things, they are very important but they are not reaching the clinical teams, who are the majority of clinical staff in the [department], and who play a huge role in implementing research, in helping research happen, in spreading the message about research to patients.* R9, doctoral

*It's been sad that my clinical NHS organisation doesn't really seem to be... promoting or engaging with, [or] even knowing about, [my research] work. And not just my work...I see that with other colleagues... and all their [research] has had no impact in how we work at the Trust level. And, yeah, that sort of, doesn't really feel right, that I'm much more known in [another continent] than in my hospital, or elsewhere than in my hospital. And that not being known is important.* R4, post-doctoral

Furthermore, it was perceived that research opportunities were not equally accessible for all NMAHPPs across the Trust. This was specifically reported among interviewees from nursing and midwifery professions, as illustrated by R12 and M5:

*Before I took this [research project] on, I was very unaware of the numbers of non-medical clinical researchers, very unaware of publications and studies that were going on. I think it's generally very under-advertised. The routes into it are not clearly defined.* R12, pre-doctoral

*We've still got a real issue with research and clinical academia, because I think it's very much a block for people. People assume that it's someone over there that's very academic and very clever, that's educated to a higher level than they are.* M5

### Clinical academic pathways

The transient nature of funding for clinical academic work was flagged as a negative impact of the current clinical academic pathway by both researchers and managers. The model in place at the NHS Trust centred on individuals applying for research funding to buy out their clinical time for a specified duration in order to complete their research project or fellowship. This raised two key concerns: what happens *when the funding ends*; and finding suitable *backfill* to support the clinical service. There was also a widespread perception among interviewees that the clinical academic pathway for doctors appeared more clearly defined, and easier to access and navigate, although no one was able to recall what this pathway entailed (medical model). Despite these reported challenges, the majority of research-active clinicians were keen to pursue further research and reported personal job satisfaction associated with their combined research and clinical roles, describing this as the best of both worlds.

### *Applying for research funding*

Research-active clinicians recalled the requirement to secure funding to enable dedicated research time. This often involved devoting their own time to complete the application and/or preliminary research, as indicated by R2:

*I was doing my applications for these fellowships and there were two of them and they were both over the same time... so out of work I was doing a lot of the study and I didn't feel like I had a lot of time for myself, and then when I turned up at work it was always crazy so didn't really have... yeah, so I think that was kind of overwhelming me a little bit.* R2 pre-doctoral

However, the large majority of research-active clinicians were keen to continue to pursue a clinical academic career and described their plans for future funding applications. It appeared that a key impact to the clinicians who had embarked on a programme of research, was a desire to continue to incorporate research into their clinical role, as illustrated by R1 and R8:

*I mean my aim will be to, as soon as possible, apply for some sort of postdoctoral funding, probably the [Trust] charity because that's probably the most obvious first step.* R1, doctoral

*My desire, and my perspective, and what I want to do, I only feel that much stronger, to be honest with you. I just need to figure out how I can make it work… I would like to do a PhD on this. So, if I do a PhD and prepare myself, I think I would bring a lot of benefit to my Trust.* R8, pre-doctoral

### When the funding stops

Interviewees also recalled the practical difficulties of returning to their clinical role at the end of each period of funding as illustrated by R6 and R1 (below). This overlaps with the sub-theme of (in)visibility and (in)accessibility discussed above.

*For me coming back from my PhD and even in the current environment, without a formal clinical academic pathway I think there's a risk that your research career is going to stall, and I think that happened for me actually immediately after my PhD. And that's a shame, isn't it?* R6, post-doctoral

*I will just go back to my previous clinical post, which is already starting to, not exactly panic me, but I don't feel like it would be the right thing. I don't feel like it would be exactly the best move for me, but equally I'll need the job and the money, so I will do it if that's the only option that's available to me.* R1, doctoral

For managers, there was no clear strategy on how to best incorporate the returning research-active clinicians' skills into their clinical role. There appeared to be a friction between a desire to embrace the positive influences on the research culture of the team, with the need to maintain clinical service outputs, as illustrated by M5 and M6:

*I mean, how best to use them? Well, I suppose first of all, in a pragmatic way, it's about honouring the fact that they've got this knowledge, they've done this piece of research, so it's how we are using that research to change the service. There's something about them coming back in and honouring their achievement, so, actually should they get paid more? Because this is the trouble. If you don't honour them from that perspective, they will go on and be, you know, go into another organisation.* M5

*It's difficult because you then have to come back to a job, and you've stayed static and others have progressed, so you're going to have to drop back down to where you were… You know, having got their PhDs is fantastic, but then we're struggling because we have a clinical service to run that we can't, you know, I can't give them a post… If they could slot nicely into a clinical academic post that's funded, that would be fantastic! The trouble is it's so difficult… we just need them to be working [clinically]."* M6

### Backfill chain

Most interviewees recalled that a major impact of clinical academic activity, was the need to secure backfill to cover clinical time/duties. This was perceived as being time consuming and creating operational difficulties for managers, particularly if the research fellowships were part-time, or within small departments, as summarised by M3 and M8:

*The frustration of backfill is getting comparable people to cover the gap… So you kind of have to accept you might have a gap in the service.* M3

*It's challenging because often, they are… if it's a full-time fellowship, that can be sometimes easier, but what often happens is they're part-time, and that creates a back-fill chain, because the people that are taking fellowships are quite senior, so part of the post becomes available, somebody, junior to them applies and often is successful, so it creates this back-fill chain.* M8

However M7, who was responsible for a team of >400 clinicians from a single discipline, recalled that they were able to offer greater flexibility with backfill, due to the size of the department and the nature of the shift pattern:

*With shifts, it's really flexible, so that absolutely wouldn't be a problem, and with people that go off 50% of the time to pursue something different, we just cover them. So, it'd be like they'd have a clinical job share… we've got such a big team, so, actually, losing a few [clinicians to research fellowships], it doesn't have such an impact.* M7

Interviewees described that 'creative thinking' (M4) was required to piece together 'a jigsaw' (M8) of the necessary backfill. For example, by increasing working hours or downgrading the post:

*In a weird way I backfilled myself for one day of it because I was only 0.8 so then I went full time and needed 0.2 of it and then the other 0.2 we got backfill for.* R11, pre-doctoral

*Right now in my post, although I'm 0.5, the 0.5 of my salary is about 0.7 of a [lower clinical grade], so they should be time rich for patient care. So you get more for your money with backfilling us, if you look at it that way.* R3, post-doctoral

Others were concerned that downgrading might have a detrimental effect of the long-term sustainability of both research fellowships and clinical posts, especially given extensive financial pressures. For example, the risk that clinical posts could become permanently downgraded, potentially undermining the value of the service, and that of the research-active clinicians:

*It centres around the research culture and an understanding of the value of research and then on a very practical level just getting support to either undertake a fellowship and get the right and appropriate backfill. I mean, I've been in a situation where the backfill for my post has actually been downgraded because it's seen as an opportunity to perhaps save some money for the Trust, and that's not sending the right message, is it?* R6, post-doctoral

### Medical model

There was a widespread perception that involvement in clinical academic activity was more accessible for doctors compared with NMAHPPs, as discussed by R11 and M1:

*I think the problem for us as non-medics is that this type of research is not within our career progression… people aren't*

*coming at you like they do with the medics and saying, 'Okay you're a [junior doctor] and now you've got to do your research otherwise you're not going to get a consultant post'. It's completely different. And, so because of that, I found it really challenging to access any kind of help and support initially.* R11, pre-doctoral

*My perception is the medics have got it in the can… but I couldn't tell you what that really means. I do know there's probably more of an acceptance or an expectation perhaps that medics do some research.* M1

It appeared that many of the perceived detrimental impacts described above were associated with the absence of a clear clinical academic career pathway for NMAHPPs, several interviewees were concerned that it would not be appropriate to simply replicate the existing model for medics, as illustrated by R5 and M2:

*The medics have had that history of those two things [research and clinical practice] being intertwined more from a longer time. For [NMAHPPs] it isn't as established, so we're in different territory. And whether we should be trying to head the same way or cultivate entirely new arrangements is another question, but [we] just don't have that track record.* R5, pre-doctoral

*I also think, sadly, that in some of the medical staff, there is a snobbery, and there is a snobbery that, actually, only medics do research, and only the research that medics do is valuable and valid, and all the rest of it.* M2

### Best of both worlds

Despite the challenges described above, the perception among research-active clinicians appeared to be one of enhanced job satisfaction through the combination of clinical and research roles, as illustrated by R9 and R10. This emerged as the driving force for pursuing clinical academic opportunities, and was also highlighted by the majority of managers, as described by M2 and M7 (below):

*I would rather be a clinical academic than an academic, because it is what gives me the butterflies. It's basically this combination between having hands-on the clinical reality and then based on the questions that… are experienced by me in the clinical practice, having the privilege to be given the time to go and try to answer them, and then give back the results and the benefits of those answers to the clinical environment. That is what really motivates me.* R9, doctoral

*Far greater career satisfaction comes from a more varied role… And having the tools to impact care at a deeper and wider level beyond the day-to-day level of clinical provision.* R10, post-doctoral

*It allows you to combine the best bits of what they want they do into one job, because… I think, looking at the people that we've got in the clinical academic roles, it allows them to get the best of both worlds for what they want.* M2

*I think just giving people opportunities to pursue different avenues, it makes them more motivated in-work, as well. So,*

*we know that happy staff provide better patient outcomes, we absolutely know that.* M7

### Making impact tangible

All interviewees were asked a general question: how do you think we can best capture and report the impact of clinical academic activity at the Trust? This prompted a range of different responses exploring which impacts were considered important, as well as what could be captured and reported. Responses were broadly categorised as: direct research outputs, indirect research outputs, workforce impacts and patient impacts. These categories largely represented the first two themes identified above, with a focus on capturing the impacts that bring positive visibility and cultural change within the Trust.

### *Direct research outputs*

Direct research outputs included metrics that are typically required in fellowship reports, such as publications and conference presentations. While these were perceived as quantifiable measures, there was also an appreciation of quality, such as the impact factor and reach of the journal, and scope and audience of the conference. However, the large majority of interviewees suggested that a better method of measuring research impact might be to explore the implementation of the research findings, as illustrated by R12 and M4:

*'These were the results of my study, we've changed our practice and here are the opinions of the person who did it, the ward manager for example and this is how things have changed'… Those would be the kind of statements that have the most weight, in my opinion. This is the positive change that can come about through these kinds of fellowships.* R12, pre-doctoral

*I mean any kind of change in practice that's based on research that's done that's funded by the college or an academic funder. That's what we should focus on. My version, my world of the NHS doesn't care about number of publications or where you publish or how many talks you've given. That's not an important driver outcome.* M4

### *Indirect research outputs*

Indirect research outputs focused on the contributions of the research-active clinicians to the development of their profession or clinical specialty. At a national level, this included establishing treatment guidelines, involvement in professional bodies and peer reviewing, as illustrated by R6. Securing funding for additional research projects or clinical services, and sharing learning with the clinical team were also perceived as important impacts locally, as suggested by R10. However, it was acknowledged by several managers that research-active clinicians contributed their own time for many of these activities, as reported by M1.

*I've been able to influence national [clinical discipline] policy and change practice. So, I mean, that's… it's been very satisfying to see that and to know that that's happened in an evidence-based way… I think we need a way of measuring,*

*you know, if you've been asked to contribute to a national guideline. Is that being recorded somewhere and is that being recognised?* R6, post-doctoral

*[Research-activity] leads to further funding or other projects, or supporting others to do work in the field.* R10, post-doctoral

*They're being asked to speak, they're chairing panels that's all time, and it's rare that the academic PAs [programmed activities] in their job plan fully support all of that activity. And I know from personal experience that people spend an awful lot of their additional own time preparing for things… I think, there's an acceptance people will do a lot in their own time.* M1

### Workforce impacts

Proposed workforce impacts included improved recruitment and retention of staff, as highlighted in the subtheme *positive reputation* (above), and increased involvement of clinical staff in research, as illustrated by R8 and M8:

*I think [the] evidence is – each individual department, you can see how many [NMAHPPs] are involved in research, what they're doing, if they have funding permanently, if they are supported – let's say one day a week… So, all of this is evidence that your department is active academically as well as the clinical that is going on.* R8, pre-doctoral

*Even if somebody's come into the service and hasn't had an interest in research, then, you know, it grows their interest in it, it helps them go down the path.* M8

### Impacts to patients

Interviewees highlighted the importance of capturing changes in patient outcomes that were associated with research activity, however it was acknowledged that associations between research activity and patient outcomes might be difficult to identify, as noted by R1. Other suggestions included feedback from patients and Patient and Public Involvement (PPI) representatives:

*The broader NHS or Health Education England have acknowledged that a research active trust has better outcomes for patients, there must be a way of tracking that, that if you can show a difference year on year in terms of the research activity of your clinicians compared to the kind of outcomes of patients, but obviously that's a massive piece of work to stratify it… On a really macro level: 'Here's how many people are research active, here's how many patients are having good outcomes' and then, but also on a more micro level, project by project.* R1, doctoral

*I think the patient voice has to be the most powerful doesn't it, the impact on patients and directly linking it to that work.* M1

### DISCUSSION

Our qualitative exploration of the perceived impacts of clinical academic activity among NMAHPPs described these impacts across four themes. The first theme

described cultural changes including beneficial shifts in patient care and research culture. The second theme explored visibility. Clinical academic activity was believed to generate a positive reputation for the organisation, however, there were also perceived elements of invisibility and inaccessibility of clinical academic roles. The third theme discussed the impacts of the clinical academic pathways, including the precarity of clinical academic roles and the associated challenges for individuals and clinical teams. The final theme highlighted possible methods of capturing and reporting these impacts.

Perceived impacts of clinical academic activity were largely positive and focused either directly on the generation of evidence and the delivery of evidence-based care, or indirectly via expanding research awareness and providing research support within clinical teams. Similar attributes have been reported following the introduction of specific research fellowships,[34 35] interventions to increase research activity among clinical staff[36 37] and research practitioner roles.[38 39] In the current study, research activity was also associated with increased self-confidence in discussing the available evidence with patients and involving patients in shared clinical decision making. Person-centred care and shared decision making are characteristics that healthcare systems strive for,[40 41] and our findings suggest that research-active clinicians are well placed to support patients' and clinicians' understanding of the available evidence to enable informed decision making. Interviewees emphasised that assessments of research impact should aim to capture these aspects of care delivery, rather than the current perceived focus on academic outputs, such as publications, however it was acknowledged that this may be difficult in practice. Published approaches to support and measure research translation and impact within healthcare, include prospective implementation plans with clearly identified outcomes and use of implementation reporting guidelines.[11 42] These strategies may be valuable when exploring the broader impacts of clinical academic activity, although attributing recorded changes to an individual study remains challenging.

Most managers named individual research-active clinicians within their teams and highlighted beneficial outcomes in terms of service delivery and research engagement. Research-active clinicians were labelled as 'pied pipers' (M1) and drivers of change. However, there were cautionary suggestions that research engagement was driven by, and often dependent on, these individuals, and was not necessarily fully embedded in the service. A recent rapid review of theoretical frameworks for embedding research culture into allied health practice suggested that a sustainable change requires four factors: (1) organisational structures, policies and governance that support and value evidence-based practice; (2) research capability and advocacy among healthcare managers and leaders; (3) dedicated research positions, time allocated to research, and access to education and research infrastructure and (4) individual research skills, capabilities and motivation.[43]

Interviewees in the current study described the positive impacts of developing individual skills, capabilities and motivations, however, they also illustrated that the structures to support research, including protected time and recognition, were not well embedded within the organisation. For managers, there appeared to be a conflict between wanting to support and enable the development of research-active clinicians, anticipating the beneficial effects this might have, versus needing sufficient staffing to provide a clinical service. Our findings suggest that the value of clinical academic research posts for the service have not yet been realised and that the tension between 'research' and 'clinical' time still exists. The development of a sustainable research culture requires dedicated research positions, protected research time and a clinical academic career structure, which has proved challenging across international settings.[44 45] Some organisations have shown that it is possible to develop bespoke solutions to ensure career progression.[46]

The second theme of visibility revealed the role that individual research-active clinicians have in developing research capability and motivating their clinical team. Areas without clinical academic role models may not have the same level of exposure, or encouragement, to pursue research opportunities across the organisation. Without a personal connection to these activities, staff may perceive them as inaccessible. Appropriate NMAHPP role models have previously been identified as enablers of research activity,[47] and interviewees in this study gave examples of where this had occurred. However, although interviewees suggested that research engagement should be captured through quantitative data on the number of registered projects (audit, quality, improvement and research) and individuals involved in these projects, no one mentioned using existing measures of research awareness/engagement such as research spider or research culture and capacity survey, which have been reported in other NMAHPP clinical academic contexts.[48–50] It is possible that measures of actual research activity held greater importance for interviewees compared with more abstract measures of research knowledge and research intention; however, it is also possible that interviewees were not aware of the existing survey measures.

Traditional research impact metrics, such as publications and presentations were highlighted by all interviewees as a means of recording research outputs, however little value was attributed to these activities in isolation. They appeared to be seen as a step towards the introduction of new evidence into practice, while also contributing to the development of a positive reputation for the individual, their team and the organisation as a whole. This aligns with the San Francisco Declaration on Research Assessment, which calls for increased emphasis on research outputs, such as the creation of data sets and influence on policy and practice, instead of publication counts and journal Impact Factor.[51]

When interviewees were asked about the value of different impacts, sustained opportunities to be involved in research were highlighted as a factor that might improve staff recruitment and retention, and recruitment data was suggested as another method of capturing the impact of clinical academic activity. Similar views have been reported elsewhere.[38] However, while research involvement was seen as a positive driver for the workforce, the process of applying for and securing research funding was also seen as challenging for both individuals and teams. A recent mixed-methods study of NMAHPP clinical academic careers recommended investment in clinical academic roles to enable the continued utilisation of research-active clinicians' skills and experience.[47] In our study, the clinical academic model for doctors was perceived as a clearer and more established pathway with dedicated clinical academic positions; however, there were concerns that it might not be appropriate, or possible, to directly emulate this pathway due to differences in clinical roles and level of postgraduate clinical experience.

We attempted to use an inclusive definition of healthcare professions outside medicine to reflect the clinical academic strategy within the Trust. The term NMAHPP is not universally adopted, and therefore, we also used the description 'outside medicine'. This, along with similar terms, such as 'non-medical', may hinder the establishment of a distinctive clinical academic identity for this broad group of clinicians. While there is growing international interest in the development of sustainable NMAHPP clinical academic careers, current job descriptions and pathways vary and there are few substantive posts.[46 52] A universally adopted term to describe these clinicians, ideally without focusing on the fact that they are not clinical doctors, may aid the collaborative development of these roles across the different professional disciplines.

Comparison of the research impacts reported in the current study with those in existing frameworks highlights interesting contextual differences. Key themes identified in a recent systematic review of methodological frameworks for impact assessment in healthcare research focused on the macrolevel, for example, influence on policy making and health-related and societal impact.[8] In contrast, interviewees in the current study gave their perspectives of the impacts of the clinical academic activity on the day-to-day delivery of care and the skills and expertise available within their team/department. This may reflect the fact that most of our clinical academic interviewees were early career researchers and had not yet explored impacts beyond the local context. It also illustrates how a broad range of relevant stakeholders will need to be involved in determining local, national or international assessments of research impact.[10 11] Another difference between our study and existing methodological frameworks was our exploration of the impact of clinical academic activity, rather than research per se. For our interviewees, the process of NMAHPP clinicians getting to a position to be able to conduct research (ie, securing funding and backfill), and the impact of this on their

team were also essential components, which needed to be repeated for each new research study or fellowship.

Our recent systematic review explored the impacts of NMAHPP clinical academic activity reported in the literature and used the VICTOR framework to categorise the identified impacts.[13] Across all categories, there were three recurring sub-themes: the challenges and benefits of balancing clinical and academic roles; creation and implementation of new evidence; and collaboration and networks. The first two of these subthemes were also reflected in the current study, suggesting that these are likely to be important features in developing clinical academic careers and areas where impact could be assessed. We believe the ability to develop and use collaborations and networks is dependent on securing a clinical academic career structure within individual organisations, and that investment is required to ensure that clinical academics are in a position to progress beyond one-off fellowships. An ideal clinical academic pathway would include opportunities at all clinical grades, with common pathways available for all disciplines. This would enable protected time for research, dissemination and implementation activities, reducing the need for short periods of backfill, while developing future clinical and research leadership.

## Limitations

The current evaluation was conducted within a large, multisite NHS organisation and an important limitation is that the findings may not represent different healthcare environments or geographical settings. However, the comparison of our findings with the existing literature suggests that similar themes are likely to be important elsewhere. Our study was not restricted to the evaluation of one specific type of research fellowship, or other intervention, and therefore, reflects different clinical academic scenarios that occur within the NHS. Clinical academic activity was defined as engagement in research alongside clinical practice that was supported by additional funding. We acknowledge that other service development and quality improvement activity occurs within the organisation, but we were guided by the Health Research Association definition of research, and therefore, did not include activities defined by the former two categories.[24]

It is possible that interviewees may have responded in a way that they felt was socially desirable, however, steps were taken to facilitate open dialogue and explore both positive and negative aspects of clinical academic activity. Strategies included an interviewer who was not known to the interviewees in their clinical or research settings (they work clinically at a different NHS organisation), and the opportunity for interviewees to review their transcripts to ensure appropriate anonymity. Inviting interviewees to review their transcript and contribute to data interpretation does raise the potential issue that meanings expressed during the interviews may be modified as part of this process. In reality, member checking resulted in minimal changes to the written transcripts and instead

provided additional context with interviewees clarifying meaning that aided data interpretation. Involvement of interviewees and the wider research-active community with the data analysis also appeared to contribute to the on-going development of research collegiality among NMAHPPs at the Trust.

The research team comprised clinical academics from nursing and physiotherapy. To ensure that study development was informed by a broader range of disciplines, interviewees and other research-active clinicians were included in prepiloting and refining the interview schedule, reviewing and developing the framework analysis and resulting themes/subthemes to minimise the influence of the study team.

It was interesting that the views of managers and research-active clinicians were well aligned. Previous research has identified non-facilitating managers at organisational and local levels as key barriers to the development of clinical academic opportunities,[45 53 54] and this was a problem experienced by some of our clinical academics. However, interviewees were a self-selected population who responded to email invitations to discuss research activity and it is possible that the managers who participated held more positive views towards clinical academic activity than others within and outside the organisation. Managers and research-active clinicians were identified using a purposive sampling strategy aimed at including a breadth of different experiences (clinical discipline, academic level, clinical grade, hospital site). We did not include ethnicity as a sampling criterion, nor collect ethnicity data for participants, and acknowledge this as a limitation.

Due to the relatively small number of interviewees, we were unable to fully explore potential differences in views across different clinical disciplines, for example, allied health professionals compared with nurses and midwives. However, during the analysis, we deliberately looked for possible divergent views as a means of ensuring that our themes were fully representative of the data. We were unable to identify any clear patterns, but larger samples, specifically designed to explore the question of divergent views, may uncover important differences between professions.

## CONCLUSIONS

Perceived positive impacts of NMAHPP clinical academic activity focused on interlinked positive changes for patients and clinical teams. The perception was that for patients, this included access to evidence-based treatment and evidence-informed shared clinical decision making. For clinical teams, this was experienced through positive changes to the local research culture. The availability of, and support for, research opportunities, were believed to improve staff recruitment and retention within research-active departments. However, these impacts centred around individual research-active clinicians and did not necessarily translate to all areas within the organisation.

Moreover, the internal visibility of clinical academics was often limited. The current clinical academic pathway was identified as creating challenges for managers due to a tension between supporting externally funded research-time and having sufficient staffing to cover the clinical service. Our findings suggest that the local impacts of clinical academic activity are important to individuals and to the organisation, but that sustained investment and support are required to ensure that research-active clinicians are able to realise the broad range of positive impacts identified here. It is also important that mechanisms of capturing and recording different impacts are used, so that the value of clinical academic activity is visible.

**Acknowledgements** The authors wish to thank and acknowledge all participants for giving up their time to be interviewed. Thank you also to the participants and members of the Imperial College Healthcare NHS Trust Postgraduate Research Forum for their helpful feedback and contribution to the development of the reported themes.

**Contributors** All authors contributed to the study conception and design. LN conducted the interviews, led the analysis and drafted the manuscript. CMA and MW provided input to the analysis, agreed the final themes and contributed to the content of the manuscript. All authors have approved the final manuscript.

**Funding** This study was funded by the NIHR Imperial Biomedical Research Centre (BRC).

**Disclaimer** The views expressed are those of the authors and not necessarily those of the NIHR or the Department of Health and Social Care.

**Competing interests** None declared.

**Patient consent for publication** Not applicable.

**Ethics approval** Approval was granted by Imperial College Healthcare NHS Trust Clinical Audit and Service Evaluation Team (reference: 418). The study was exempt from NHS ethics approval because interviewees were recruited by virtue of their professional role.

**Provenance and peer review** Not commissioned; externally peer reviewed.

**Data availability statement** Data are available on reasonable request. The study protocol is available from the OSF repository: DOI 10.17605/OSF.IO/P8HYD. Participants did not provide consent for open access publication of their interview transcripts. These data are available on reasonable request and subject to informed consent from the study interviewees.

**ORCID iDs**
Lisa Newington http://orcid.org/0000-0001-6954-2981
Caroline M Alexander http://orcid.org/0000-0003-1816-8939
Mary Wells http://orcid.org/0000-0001-5789-2773

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
