## [Reviewer comments · BMJ Open]

ARTICLE DETAILS

TITLE (PROVISIONAL)	The impacts of clinical academic activity: Qualitative interviews with healthcare managers and research-active nurses, midwives, allied health professionals and pharmacists
AUTHORS	Newington, Lisa; Alexander, Caroline; Wells, Mary

VERSION 1 – REVIEW

REVIEWER	Catherine Henshall Oxford Brookes University, Health and Life Sciences
REVIEW RETURNED	12-May-2021

GENERAL COMMENTS	Thank you for the opportunity to review this manuscript reporting on an important topic. It is a well written paper and I have included a few suggestions for improvement here: Comments: Abstract I would remove the sub-heading 'outcomes' as it doesn't really fit with the chosen study design Introduction Line 68: should this read systematic review? I think the Introduction would benefit from a little more literature relating to the benefits of clinical academics in terms of the impact of patient care and health outcomes. You touch on this on line 61, but more detail would add context here. It might also be worth mentioning in your Introduction some of the capacity building programmes that have been developed by the NIHR such as the 70@70 Programme and/or the Nursing and Midwifery Incubator, which is aimed at developing opportunities for NMs to pursue clinical academic developmental opportunities. Patient and public involvement Although no patients were involved, it would be useful to state if any other stakeholders were involved i.e. you mention in the section above that topic guides were piloted with research active clinicians. This information might sit better under the PPI heading. Participants and recruitment Please provide more information in this section about how you accessed and recruited the participants. Who did you seek permission to access them from and how did you approach staff? I see some of this information is in Table 1 but alluding to this in the text would be helpful. Also, it sounds from Table 1 that as well as purposive sampling you also used snowball sampling techniques as you recruited managers at the suggestion of other participants that you were interviewing?
--

	Lines 110-112: were the interviews conducted face to face or remotely? Useful to add this detail to main text as well as Table 1. Line 111: 'lead author' should read 'lead researcher' Data analysis Line 119: though the data was managed using the Framework Approach, I presume that the raw data was thematically analysed? If so, this needs to be stated. Thematic analysis is the method used and Framework is the approach used to support the analysis. Line 125-127: can you provide an example of some of the sort of feedback that was fed back to you as a result of the member checking process? Results Line 355: It's not entirely clear here to me why backfilling with a lower grade would be detrimental to the sustainability of research and clinical posts. Do you mean because then there won't be a higher banded position for the researcher to come back to at the end of their fellowship? If so, could this be made clearer? Line 419-22: This quote is in contrast to some of the previous data which cites the importance of profile raising and standing in order to influence and drive change. Publications and talks do contribute to this raised profile and enable to evidence based findings to drive forward changes in practice locally and nationally etc...this might be worth commenting on further in your Discussion section. Discussion Lines 523-535: Do you have any suggestions as to what an alternative CA model for non-medics might look like, given the differences in clinical roles and career structure? Is it possible to comment on any differences in findings across the NMAHPP participants, broken down by profession? Were there any notable similarities and/or differences between professional disciplines? Limitations: Line 572-3: was the interviewer(s) a healthcare professional? Line 573-4: it might be worth highlighting the challenges of the member checking process in terms of participants changing in terms of their interpretation of the data and the meaning assigned to it (i.e. what they felt or recalled at the time of the interviews may be different to the present day when they are reviewing the data). Figure 1: I'm not sure that this figure adds anything or at least it is not clear how the three main themes directly relate (via the arrow) to the impact outputs.
--	---

REVIEWER	Débora de Faria-Schützer Universidade Estadual de Campinas
REVIEW RETURNED	14-May-2021

GENERAL COMMENTS	I would like to thank you for the opportunity to review this manuscript. There is a great rigor in your methodology and it gives a good quality to the study. The manuscript is relevant and brings important findings and discussion. However the manuscript needs some adjustments in its organization. Following are my comments: -Patient and public involvement "The focus of this service evaluation was on understanding the perceptions of healthcare managers and clinicians and no patient/public advisors were involved."
--

	I suggest putting it in two sentences or justifying why not to include physicians and patients -Participants and recruitment "Clinical academic activity was defined as engagement in research alongside clinical practice that was supported by additional funding from clinical research organizations or charities. This included both full and part-time research secondments. Eligible disciplines were: nursing; midwifery; the allied health professions (art therapy, dietetics, drama therapy, music therapy, occupational therapy, orthoptics, operating department practitioners, osteopathy, podiatry, prosthetics and orthotics, paramedics, physiotherapy, radiography, and speech and language therapy); healthcare science and pharmacy. This was abbreviated to NMAHPPs. I think that definition can appear right at the beginning of the introduction. - I missed a bigger description of the research setting in the method. How many people were invited to participate in the survey? How many refused? - The table 1 is very good and beautiful, but it will get more didactic in the article presenting it in two separate tables (one in the methods on recruitment and the other in the results with the data on the participants). - Results are consistent and there are important findings. I think that the themes Visibility and subthemes (In) visibility and (in) accessibility, Medical model and Best of both worlds deserve to be highlighted as well as the theme Making impact tangible. Therefore, I suggest presenting these results first in the text. - Discussion: The discussion is very interesting and consistent. I would like to share a thought that occurred to me during the reading of the manuscript: does the term "non-medical clinical researchers" not reaffirm a view centered on the doctor and not on health professionals? I also thought it would be challenging and interesting to create a figure with practical improvement tips for clinical academic activity, which your research has found.
--	--

VERSION 1 – AUTHOR RESPONSE

REVIEWER 1 a) REVIEWER'S COMMENT Thank you for the opportunity to review this manuscript reporting on an important topic. It is a well written paper and I have included a few suggestions for improvement here: Abstract I would remove the sub-heading 'outcomes' as it doesn't really fit with the chosen study design AUTHORS' RESPONSE Thank you for this feedback and comment on the value of our work. We agree that 'outcomes' is not an ideal fit for this qualitative design and have renamed the heading as 'analysis'. Location: Abstract, page 2, line 37	
b) REVIEWER'S COMMENT Introduction Line 68: should this read systematic review? AUTHORS' RESPONSE Thank you for noticing this omission, yes it should read systematic review. We have amended the text. In addition, the review has subsequently been published in BMC Health Services Research and we have also updated the reference. Location: Introduction, page 3, line 71 and reference 13	

c) REVIEWER'S COMMENT

I think the Introduction would benefit from a little more literature relating to the benefits of clinical academics in terms of the impact of patient care and health outcomes. You touch on this on line 61, but more detail would add context here. It might also be worth mentioning in your Introduction some of the capacity building programmes that have been developed by the NIHR such as the 70@70 Programme and/or the Nursing and Midwifery Incubator, which is aimed at developing opportunities for NMs to pursue clinical academic developmental opportunities.

AUTHORS' RESPONSE

Thank you for this helpful suggestion. We have added additional text describing the potential benefits of clinical academic activity in relation to patient care and health outcomes. We have also referenced a number of the other research opportunities, including those you highlighted.

Additional text:

For example, research activity has been associated with improvements in organisational performance and efficiency, patient satisfaction and confidence in their healthcare professionals, and staff satisfaction [3, 6]. Research activity has also been associated with reductions in mortality and staff turnover [2, 5, 6].

Location: Introduction, page 3, lines 62-5

Additional text:

This is in addition to research leadership capacity building initiatives such as the NHS 70@70 [17], research internships for newly qualified clinicians [18], nursing, midwifery and allied health professionals research awards [19], and discipline-specific research capacity building initiatives, such as the NIHR Nursing and Midwifery Incubator [20].

Location: Introduction, page 3, lines 81-4

d) REVIEWER'S COMMENT

Patient and public involvement

Although no patients were involved, it would be useful to state if any other stakeholders were involved i.e. you mention in the section above that topic guides were piloted with research active clinicians. This information might sit better under the PPI heading.

AUTHORS' RESPONSE

Thank you for this suggestion. We have moved the information concerning clinician involvement in developing the interview topic guide to the PPI section. We have also referenced some of the PPI work that contributed to our broader programme of research. This response also relates to comment b from reviewer 2, who made similar suggestions.

Amended text:

The focus of this study was on understanding the perceptions of healthcare managers and clinicians from the professions outside medicine. Patient/public advisors were not specifically involved, however, the wider topic of research impact was discussed with two public representatives as part of our larger programme of research [26]. Interview topic guides were developed in collaboration with research-active clinicians from both within and outside the NHS Trust. Involvement included providing feedback and suggestions on the initial draft, piloting and further refining the final version. The medical and dentistry community were not included in this stakeholder work because their clinical academic careers are already well established, and our emphasis was solely on the professions outside medicine.

Location: Patient and Public Involvement, page 4, lines 100-7

e) REVIEWER'S COMMENT

Participants and recruitment

Please provide more information in this section about how you accessed and recruited the participants. Who did you seek permission to access them from and how did you approach staff? I see some of this information is in Table 1 but alluding to this in the text would be helpful. Also, it sounds from Table 1 that as well as purposive sampling you also used snowball sampling techniques as you recruited managers at the suggestion of other participants that you were interviewing?

Lines 110-112: were the interviews conducted face to face or remotely? Useful to add this detail to

main text as well as Table 1.
Line 111: 'lead author' should read 'lead researcher'

AUTHORS' RESPONSE

Thank you for highlighting these points. We have provided additional information about the recruitment process in the text, including the use of snowball sampling techniques within our purposive sampling strategy. We have also included details on the format of the interviews in the main text and amended 'lead author' to 'lead researcher'.

Amended text:

Permission to directly contact individuals was granted through the Trust Clinical Audit and Service Evaluation Team. Potential interviewees were contacted by email, using addresses that were openly accessible through the NHS or university email systems.

Location: Participants and recruitment, page 5, lines 121-4

Amended text:

A purposive sampling strategy was adopted to ensure inclusion of a range of experiences. This also included snowball sampling techniques, with interviewees and potential interviewees asked to suggest other research-active clinicians and managers.

Location: Participants and recruitment, page 5, lines 125-7

Amended text:

Interviews were conducted by the lead researcher and were delivered face-to-face or remotely, according to interviewee preference.

Location: Participants and recruitment, page 5, 129-30

f) REVIEWER'S COMMENT

Data analysis

Line 119: though the data was managed using the Framework Approach, I presume that the raw data was thematically analysed? If so, this needs to be stated. Thematic analysis is the method used and Framework is the approach used to support the analysis.

Line 125-127: can you provide an example of some of the sort of feedback that was fed back to you as a result of the member checking process?

Thank you for these suggestions. We have added that *the coded text was explored thematically* (Location: Analysis, page 6, line 145). We have also provided examples of the outcome of the member checking process.

Additional text:

Example feedback included discussion on how the individual codes could be arranged as themes and sub-themes, suggestions for the wording of the theme headings and ideas for the design of the summary model.

Location: Analysis, page 6, line 150-2

g) REVIEWER'S COMMENT

Results

Line 355: It's not entirely clear here to me why backfilling with a lower grade would be detrimental to the sustainability of research and clinical posts. Do you mean because then there won't be a higher banded position for the researcher to come back to at the end of their fellowship? If so, could this be made clearer?

AUTHORS' RESPONSE

Thank you for highlighting this lack of clarity. We have added an additional sentence to provide more context.

Additional text:

For example, the risk that clinical posts could become permanently downgraded, potentially undermining the value of the service, and that of the research-active clinicians:

Location: Backfill chain, page 14, lines 386-8

h) REVIEWER'S COMMENT

Line 419-22: This quote is in contrast to some of the previous data which cites the importance of profile raising and standing in order to influence and drive change. Publications and talks do contribute to this raised profile and enable to evidence based findings to drive forward changes in practice locally and nationally etc...this might be worth commenting on further in your Discussion section.

AUTHORS' RESPONSE

Thank you for this comment. We found it interesting that, while interviewees did see the value of publications and presentations, there was also a perceived disconnect between these particular measures of impact and direct benefits to patient care or wider research engagement. We have added an additional paragraph to explore this in the discussion.

Additional text:

Traditional research impact metrics, such as publications and presentations were highlighted by all interviewees as a means of recording research outputs, however little value was attributed to these activities in isolation. They appeared to be seen as a step towards the introduction of new evidence into practice, while also contributing to the development of a positive reputation for the individual, their team, and the organisation as a whole. This aligns with the San Francisco Declaration on Research Assessment, which calls for increased emphasis on research outputs, such as the creation of data sets and influence on policy and practice, instead of publication counts and journal Impact Factor [51].

Location: Discussion, page 22, lines 556-62

i) REVIEWER'S COMMENT

Discussion

Lines 523-535: Do you have any suggestions as to what an alternative CA model for non-medics might look like, given the differences in clinical roles and career structure?

AUTHORS' RESPONSE

Thank you for the encouragement to include our suggestions for the direction of future clinical academic pathways. We have added additional text to include the ideas that arose from our data.

Additional text:

An ideal clinical academic pathway would include opportunities at all clinical grades, with common pathways available for all disciplines. This would enable protected time for research, dissemination and implementation activities, reducing the need for short periods of backfill, while developing future clinical and research leadership.

Location: Discussion, page 23, lines 604-7

j) REVIEWERS' COMMENT

Is it possible to comment on any differences in findings across the NMAHPP participants, broken down by profession? Were there any notable similarities and/or differences between professional disciplines?

AUTHORS' RESPONSE

This is an interesting point. We deliberately looked for differences between broader clinical groups, for example nurses and midwives compared with allied health professionals as part of our analysis. This was to ensure that our themes were representative of the data and to enable divergent views to be explored and reported. There were no obvious differences or patterns between these broad clinical group, but our sample was not designed to specifically explore this issue. We have added an additional section in the limitations to discuss this.

Additional text:

Due to the relatively small number of interviewees, we were unable to fully explore potential differences in views across different clinical disciplines, for example allied health professionals compared with nurses and midwives. However, during the analysis we deliberately looked for possible divergent views as a means of ensuring that our themes were fully representative of the data. We were unable to identify any clear patterns, but larger samples, specifically designed to explore the question of divergent views, may uncover important differences between professions.

Location: Limitations, page 24-5, lines 644-9

k) REVIEWER'S COMMENT

Limitations: Line 572-3: was the interviewer(s) a healthcare professional?

AUTHORS' RESPONSE

Thank you for highlighting that this was unclear. We have added additional text to clarify that the interviewer is a healthcare professional.

Amended text:

Strategies included an interviewer who was not known to the interviewees in their clinical or research settings (they work clinically at a different NHS organisation), and the opportunity for interviewees to review their transcripts to ensure appropriate anonymity.

Location: Limitations, page 24, lines 621-3

l) REVIEWER'S COMMENT

Line 573-4: it might be worth highlighting the challenges of the member checking process in terms of participants changing in terms of their interpretation of the data and the meaning assigned to it (i.e. what they felt or recalled at the time of the interviews may be different to the present day when they are reviewing the data).

AUTHORS' RESPONSE

Thank you for highlighting this important point. In the current study, there were very few changes requested as part of this member checking. We have added an additional section in the limitations to discuss this issue.

Additional text:

Inviting interviewees to review their transcript and contribute to data interpretation does raise the potential issue that meanings expressed during the interviews may be modified as part of this process. In reality, member checking resulted in minimal changes to the written transcripts and instead provided additional context with interviewees clarifying meaning that aided data interpretation. Involvement of interviewees and the wider research-active community with the data analysis also appeared to contribute to the on-going development of research collegiality among NMAHPPs at the Trust.

Location: Limitations, page 24, lines 623-7

m) REVIEWER'S COMMENT

Figure 1: I'm not sure that this figure adds anything or at least it is not clear how the three main themes directly relate (via the arrow) to the impact outputs.

AUTHORS' COMMENT

Thank you for raising this point. Reviewer 2 also provided feedback on the figure. We have redeveloped this figure and now categorise the three themes and sub-themes under the headings 'positive impacts', 'challenges' and 'opportunities'. This also incorporates the discussion of what an ideal clinical academic pathway might look like, based on our data. We hope that this supplements our written findings with a visual illustration of the identified themes and how they interlink.

REVIEWER 2**a) REVIEWER'S COMMENT**

I would like to thank you for the opportunity to review this manuscript. There is a great rigor in your methodology and it gives a good quality to the study. The manuscript is relevant and brings important findings and discussion. However the manuscript needs some adjustments in its organization.

AUTHORS' RESPONSE

Thank you for your helpful comments and suggestions for re-structuring the manuscript. Please see our responses to your individual points below.

b) REVIEWER'S COMMENT

Patient and public involvement

"The focus of this service evaluation was on understanding the perceptions of healthcare managers and clinicians and no patient/public advisors were involved."

I suggest putting it in two sentences or justifying why not to include physicians and patients –

AUTHORS' RESPONSE

Thank you for suggesting this. Reviewer 1 also made suggestions relating to the patients and public involvement section. We have provided some additional information about the broader patient and public involvement activities that were carried out as part of our larger programme of research. To illustrate the use of stakeholder engagement activities more clearly, we have moved the information relating to clinician feedback and involvement in the development of the project into the patient and public involvement section. We have also added some additional text and references within the introduction to highlight the various clinical academic routes within medicine. And some additional text to demonstrate why physicians were not included as part of our stakeholder involvement work.

Additional/amended text:

The focus of this study was on understanding the perceptions of healthcare managers and clinicians from the professions outside medicine. Patient/public advisors were not specifically involved; however, the wider topic of research impact was discussed with two public representatives as part of our larger programme of research [26]. Interview topic guides were developed in collaboration with research-active clinicians from both within and outside the NHS Trust. Involvement included providing feedback and suggestions on the initial draft, piloting and further refining the final version. The medical and dentistry community were not included in this stakeholder work because their clinical academic careers are already well established, and our emphasis was solely on the professions outside medicine.

Location: Patient and public involvement, page 4, lines 100-7

Additional text:

Within medicine, there are various career pathways and structures to support clinical academic roles [14, 15].

Location: Introduction, page 3, line 78

c) REVIEWER'S COMMENT

Participants and recruitment "Clinical academic activity was defined as engagement in research alongside clinical practice that was supported by additional funding from clinical research organizations or charities. This included both full and part-time research secondments. Eligible disciplines were: nursing; midwifery; the allied health professions (art therapy, dietetics, drama therapy, music therapy, occupational therapy, orthoptics, operating department practitioners, osteopathy, podiatry, prosthetics and orthotics, paramedics, physiotherapy, radiography, and speech and language therapy); healthcare science and pharmacy. This was abbreviated to NMAHPPs. I think that definition can appear right at the beginning of the introduction.

AUTHORS' RESPONSE

Thank you for this suggestion. We have revised the text as follows:

Amended text:

Eligible research-active clinicians were healthcare professionals from any discipline outside medicine who worked within the NHS Trust and were engaged in clinical academic activity [28]. This included: nursing; midwifery; the allied health professions (art therapy, dietetics, drama therapy, music therapy, occupational therapy, orthoptics, operating department practitioners, osteopathy, podiatry, prosthetics and orthotics, paramedics, physiotherapy, radiography, and speech and language therapy); healthcare science, clinical psychology and pharmacy, as abbreviated to NMAHPPs. Clinical academic activity was defined as engagement in research alongside clinical practice that was supported by additional funding from clinical research organisations or charities. This included both full and part-time research secondments.

Location: Participants and recruitment, page 4, lines 111-19

d) REVIEWER'S COMMENT

I missed a bigger description of the research setting in the method. How many people were invited to

participate in the survey? How many refused?

AUTHORS' RESPONSE

Thank you for this comment. We have provided some additional contextual information about the NHS site within the methods section. The number of invitations and the number of interviews are reported in the Table, but we have now added additional information in the results section explaining the gap between the number of invitations and number of interviews (i.e. response rate).

Additional text:

Imperial College Healthcare NHS Trust is a large multi-site NHS organisation situated in north west London, UK. The Trust provides a range of specialist healthcare services located in both in- and out-patient settings, and serves around a million people per year, with a staff of >13,000.

Location: Participants and recruitment, page 4, lines 109-11

Additional text:

None of the invited managers or research active clinicians actively refused to participate, but there were two non-responders within the manager group, and four within the research-active clinician group. An additional two individuals (one in each group) were unable to schedule an interview due to changeable work commitments associated with the Covid-19 response. Three individuals responded to the Twitter invitation, although all three had already been identified as potential participants.

Location: Results, page 6, lines 155-160

e) REVIEWER'S COMMENT

The table 1 is very good and beautiful, but it will get more didactic in the article presenting it in two separate tables (one in the methods on recruitment and the other in the results with the data on the participants).

AUTHORS' RESPONSE

Thank you for your appreciation of our table design! We have now split the table as suggested to provide recruitment information to accompany the methods (Table 1), with the participants and data collection information in the results section (Table 2).

Location: Table 1 and Table 2

f) REVIEWER'S COMMENT

Results are consistent and there are important findings. I think that the themes Visibility and subthemes (In) visibility and (in) accessibility, Medical model and Best of both worlds deserve to be highlighted as well as the theme Making impact tangible. Therefore, I suggest presenting these results first in the text.

AUTHOR'S COMMENT

Thank you for the suggestion to re-order the presentation of our themes. The content of the themes and sub-themes was derived collaboratively with input from interviewees and other research-active clinicians. We therefore feel that it is important to keep the sub-themes medical model and best of both worlds within the context of the main theme 'clinical academic pathways', rather than move elsewhere within the thematic structure.

The first theme 'cultural shifts' was the dominant theme in the sense that there were few conflicting views, it is represented by all interviewees, and described the essence of what was perceived as important for clinical academics. The second theme 'visibility' explored the perceptions that while research-activity can lift the profile of the organisation, there needs to be visibility and accessibility for all, which linked into the third theme 'clinical academic pathways'. The final theme 'making impact tangible' was a composite theme derived from the discussion of how to measure/record the types of impact mentioned within the other themes. For this reason, we feel it is important that 'making impact tangible' remains as the last theme.

We would therefore prefer to keep the structure and order of the themes in the way that we, our interviewees, and the clinicians involved in the Trust research forum, believe best reflects the data.

g) REVIEWER'S COMMENT

Discussion: The discussion is very interesting and consistent. I would like to share a thought that occurred to me during the reading of the manuscript: does the term “non-medical clinical researchers” not reaffirm a view centered on the doctor and not on health professionals?

AUTHOR’S COMMENT

Thank you for highlighting this point. It is a discussion that we have also had within our research group. We prefer to use the term ‘outside medicine’ rather than ‘non-medical’ because this describes the group as something they are, rather than what they are not. However, you are right, this continues to be centred around medicine rather than all the other healthcare professions. We have added an additional point in the discussion to raise this issue.

Additional text:

We attempted to use an inclusive definition of healthcare professions outside medicine to reflect the clinical academic strategy within the Trust. The term NMAHPP is not universally adopted and therefore we also used the description ‘outside medicine’. This, along with similar terms, such as ‘non-medical’, may hinder the establishment of a distinctive clinical academic identity for this broad group of clinicians. While there is growing international interest in the development of sustainable NMAHPP clinical academic careers, current job descriptions and pathways vary and there are few substantive posts [46, 52]. A universally adopted term to describe these clinicians, ideally without focussing on the fact that they are not clinical doctors, may aid the collaborative development of these roles across the different professional disciplines.

Location: Discussion, page 22, lines 574-81

h) REVIEWER’S COMMENT

I also thought it would be challenging and interesting to create a figure with practical improvement tips for clinical academic activity, which your research has found.

AUTHORS’ COMMENT

Thank you for this suggestion. We have revised the figure and included a section labelled ‘opportunities’. This contains suggestions derived from our data for the development of clinical academic roles/careers. We have also added additional text within the discussion to link in with the amended figure, and in response to a comment from reviewer 1.

Location: Figure 1

Additional text:

An ideal clinical academic pathway would include opportunities at all clinical grades, with common pathways available for all disciplines. This would enable protected time for research, dissemination and implementation activities, reducing the need for short periods of backfill, while developing future clinical and research leadership.

Location: Discussion, page 23, lines 604-7